# Central Symptoms of Insomnia in Relation to Depression and COVID-19 Anxiety in General Population: A Network Analysis

**DOI:** 10.3390/jcm11123416

**Published:** 2022-06-14

**Authors:** Eun Jung Cha, Hong Jun Jeon, Seockhoon Chung

**Affiliations:** 1Department of Psychiatry, Konkuk University Medical Center, Konkuk University School of Medicine, Seoul 05029, Korea; eunjungcha96@gmail.com; 2Department of Psychiatry, Asan Medical Center, University of Ulsan College of Medicine, Seoul 05505, Korea

**Keywords:** depression, insomnia, anxiety, COVID-19, network analysis

## Abstract

Background: Insomnia is prevalent among the general population, and studies have shown an increase in insomnia symptoms during the novel coronavirus (COVID-19) pandemic. Despite numerous studies of insomnia, few studies have investigated insomnia symptoms in detail. In this study, we used network analysis to investigate interactions between insomnia symptoms in the general population. Furthermore, given the effect of COVID-19 on mental health, we also investigated how anxiety response to COVID-19 and depression related to insomnia symptoms. Methods: Data from 785 non-infected participants were used. The Insomnia Severity Index (ISI), Stress and Anxiety to Viral Epidemics—6 Scale (SAVE-6), and Patient Health Questionnaire—9 (PHQ-9) were used to measure insomnia symptoms, anxiety response to COVID-19, and depression, respectively. Network analysis was performed using R Studio. Centrality indices and edge weights were obtained, and each index was evaluated using bootstrapping methods. Results: The network revealed ISI7 (worry about current sleep pattern) to be the most central insomnia symptom. ISI7 was strongly connected to SAVE-6 total score, and ISI2 (difficulty staying asleep) was strongly connected to PHQ-9 total score. Conclusion: High centrality of ISI7 supports the role of dysfunctional cognitions in etiological models of insomnia and thus the cognitive behavioral therapy for insomnia. The relationship between ISI7 and SAVE-6 is explained by transposition of worry and fear of contracting COVID-19 to worry about sleep patterns. The link between ISI2 and PHQ-9 necessitate further investigations of whether specific symptoms of insomnia are more associated with depression.

## 1. Introduction

Insomnia is an important public health issue due to its symptoms, complications, and prevalence. It is characterized by a range of sleep-related symptoms, including difficulty in initiating and maintaining sleep, waking up too early, and worsened daily functioning (e.g., social, occupational) due to insufficient sleep. Insomnia symptoms are commonly experienced, and around one-third of the general population experience insomnia symptoms [1]. In an epidemiology study done in Europe, this prevalence was reported to be as common as 27.2% [2]. In addition, one study reported that 28% of insomnia patients also had current diagnosis of mental disorder [3]. Despite its prevalence and complex relationship with other mental disorders, few insomnia sufferers seek professional help, contributing to underdiagnosis and under-treatment [1,4].

Ever since the outbreak of the novel coronavirus (COVID-19), the impact of a viral pandemic on mental health has been repeatedly reported. Commonly reported effects include increased symptoms of depression, anxiety, and thoughts of suicide [5,6,7]. In addition to these symptoms, a number of studies have reported increased insomnia symptoms after COVID-19 outbreak [8,9]. Such increase was associated with comorbid mental illness, COVID-19-related stress, increased anxiety and depression, prevention measures for infection (e.g., isolation from social distancing), stigma of being infected, and negative socioeconomic effects, such as job loss [5,6,9,10]. Insomnia is an important marker in evaluating the psychological state of an individual. Studies have shown insomnia to be a significant indicator in predicting suicidal ideation [11]. In the context of COVID-19, insomnia severity fully mediated the effect of COVID-19-related fears on suicidal ideation [12]. These results warrant further research into presentations of insomnia symptoms as well as their relationships with other psychological effects of COVID-19.

Network analysis is a statistical technique that allows us to study symptom interactions in detail. It conceptualizes a disorder as a network of interacting symptoms that reinforce each other in varying degrees [13]. In a network, each symptom becomes a node, represented as circles, and their relationship with other nodes are edges, represented as lines connecting the nodes. In a network, proximal nodes indicate closer relationship, allowing easy identification of symptoms that have large and immediate effect on others. Network analysis also produces node centrality indices, which provide many perspectives in understanding of the role of each node within the network. Typically analyzed centrality indices include strength, closeness, and betweenness, which correspond to how strongly a node is related to another node, how closely a node is positioned to other nodes, and how many times a node appears in the shortest path from one node to another, respectively [14,15]. Each index holds different meanings and thus different clinical implications, allowing for detailed symptom-level analysis [15,16]. Constructing a symptom network holds multiple benefits for providing insight into the target mental disorder, and many studies have reported network analyses of various mental disorders, such as depression [17], generalized anxiety disorder [18], and panic disorder [19]. For example, a symptom with high closeness may become an intervention target, as reducing the symptom would likely lead to reductions in most other symptoms within the network. To illustrate, a study comparing depression symptom networks between those with persistent depression and those in remission showed higher closeness in feeling guilty. This suggests that feeling guilty may be a core target symptom of intervention [20]. However, to our knowledge, only a few studies have studied insomnia symptoms under the network perspective [21,22]. Furthermore, there has been no study that investigated an insomnia symptom network under the COVID-19 pandemic situation in the general population.

Therefore, the aims of this study are twofold. First, we aimed to investigate the central symptoms in the insomnia symptom network in the general population. Second, given the effect of COVID-19 on mental health, we also aimed to see how depression and anxiety response to COVID-19 relate to insomnia symptoms.

## 2. Methods

### 2.1. Study Design and Participants

Study data were derived from surveys conducted in South Korea among the general population to assess an individual’s viral anxiety. Our studies were conducted by an anonymous online survey for 14 million general population panels registered in a surveying system developed by the professional research company, EMBRAIN (www.embrain.com, accessed on 1 February 2022). The first study (study 1: 9–15 November 2021) and the second study (study 2: 10–18 January 2022) gathered 400 participants each. Among the 800 participants, 15 participants who responded ‘yes’ to having been infected with COVID-19 were excluded from the analysis, leaving data from 785 participants in total (see Appendix A). Participants voluntarily completed the survey, and we collected information about their age, sex, living region, responses to questions on COVID-19, including “Did you experience being quarantined for having been infected with COVID-19?” or “Did you experience being infected with COVID-19?”, and rating scales, including the Stress and Anxiety to Viral Epidemics—6 Items Scale (SAVE-6), Patient Health Questionnaire-9 (PHQ-9), and Insomnia Severity Index (ISI) were collected.

The Checklist for Reporting Results of Internet e-Surveys (CHERRIES) [23] was used to design the e-survey forms, and the investigator (S.C) checked the questionnaires for usability and technical functionality before implementation. The survey company provided investigators with the data only after removing personally identifiable information. The study protocols were approved by the Institutional Review Board of Asan Medical Center, Seoul, South Korea (study 1: 2021– 1490, study 2: 2021– 1755), and obtaining written informed consent was waived.

### 2.2. Measures

#### 2.2.1. Insomnia Severity Index (ISI)

Insomnia symptoms were measured using ISI [24]. ISI is a brief self-report questionnaire assessing the severity of various insomnia symptoms. It consists of seven items measuring difficulty falling asleep, difficulty staying asleep, problems with waking up too early, overall satisfaction with current sleep pattern, interference of sleep problems with daily functioning, noticeability of impaired quality of life caused by sleep problems, and worry about current sleep pattern. Items are rated using a 5-point Likert scale, ranging from 0 to 4. Higher score indicates higher severity of insomnia. The Korean version of ISI was translated and validated by Cho et al. [25]. The internal consistency reported by Cronbach’s α from this study was 0.83.

#### 2.2.2. Stress and Anxiety to Viral Epidemics—6 Items Scale (SAVE-6)

The SAVE-6 scale consists of six items that measure anxiety response to viral epidemic [26], such as fear of indefinite continuation of the virus or worry about infecting family and friends. It was derived from the SAVE-9 scale which was developed for measuring work-related stress and anxiety of healthcare workers in response to COVID-19 [27]. The SAVE-9 has two subscales of six-item viral anxiety and 3-item work-related stress subscales [28]. We developed the 6-item viral anxiety subscale as the SAVE-6 scale which can measure one’s viral anxiety among the general population. It was validated and showed good internal consistencies in various populations [29,30,31,32]. Items are rated on a five-point Likert scale, ranging from 0 to 4. Higher score indicates higher stress response to the viral epidemic. For this study, the total sum score of SAVE-6 was calculated to evaluate the overall level of stress response. The internal consistency reported by Cronbach’s α from this study was 0.83.

#### 2.2.3. Patient Health Questionnaire—9 (PHQ-9)

Depression symptoms were assessed using the PHQ-9 scale. PHQ-9 is a self-report scale consisting of nine items, in which each item measures severity of a depression symptom, such as difficulty concentrating or feeling down and depressed [33]. Each item is rated on a four-point Likert scale, ranging from 0 to 3. Total sum scores are calculated to assess severity of depression symptoms. The Korean version of PHQ-9 used in this study was translated and validated by Han et al. (2008) [34]. The internal consistency reported by Cronbach’s α was 0.91.

### 2.3. Statistical Analysis

Descriptive statistics were computed using IBM SPSS Statistics for Windows, version 27.0 (IBM Corp., Armonk, NY, USA). Means and standard deviations of age and total scores of ISI, SAVE-6, and PHQ-9 were calculated. Network analysis was performed using the qgraph package in R [35]. To perform the network analysis, total scores of SAVE-6 and PHQ-9 were divided into quartiles, in which the highest quartile was ranked 4 and the lowest quartile was ranked 1. This was done to convert the scores into ordinal scores to produce the polychoric correlation matrix required for constructing a network. The resulting network contains a total of nine nodes, in which seven nodes represent each of the ISI items and two nodes represent the sum score quartile of SAVE-6 and PHQ-9. The Fruchterman-Reingold algorithm was applied using the ‘spring’ layout, allowing strongly connected nodes to be placed closer to each other [36].

To evaluate centrality indices of each node, correlation stability coefficients (CS-coefficients) were first calculated using the bootnet package [16]. CS-coefficient is defined as the maximum proportion of samples that can be dropped to maintain a certain level of correlation between the centrality value of the sample and the centrality value of subsets obtained from bootstrapping at a 95% probability. Typically, the correlation is set to be 0.7 or higher, and a CS-coefficient larger than 0.25 is considered to have acceptable interpretability. In addition, edges were bootstrapped with 1000 samples to determine their accuracies in relation to other edges. To evaluate the significance of differences between edges, a bootstrapped difference test was also performed. The bootstrapped difference test is a null-hypothesis test to see whether each set of edge-weights significantly differ when compared with other edge-weights [16].

## 3. Results

### 3.1. Descriptive Statistics

Of the 785 participants, 400 (51.0%) were males and 385 (49.0%) were females. Participants’ mean age was 32.20 (*SD* = 12.11). The mean scores of ISI, SAVE-6, and PHQ-9 were 11.06 (*SD* = 5.39), 14.96 (*SD* = 4.31), and 5.59 (*SD* = 5.63), respectively. Results for descriptive statistics are summarized in Table 1. Total scores of SAVE-6 and PHQ-9 were divided into quartiles, and each quartile was assigned a rank ranging from 1 (first quartile) to 4 (last quartile).

### 3.2. Network Analysis

The estimated network structure consisting of ISI symptoms, SAVE-6 sum score, and PHQ-9 sum score is illustrated in Figure 1. CS-coefficients for strength, closeness, and betweenness centralities were 0.75, 0.67, and 0.28, respectively, indicating acceptable stabilities and thus allowing interpretations for all indices. However, betweenness was interpreted with care as its CS-coefficient was comparable to 0.25. Figure 2 illustrates strength, closeness, and betweenness centralities of each node. Of note, ISI7 (worry about current sleep pattern) had the highest closeness and betweenness values. For strength, ISI2 (difficulty staying asleep) had the highest value, followed by ISI7 and ISI4 (satisfaction about current sleep pattern).

Figure 3 illustrates bootstrapped edges, showing that 95% confidence intervals for a number of edges exceeded 0, suggesting significant results. The results for bootstrapped difference tests are summarized in Figure 4. Edge weights that are significantly stronger than most other edges included ISI5—ISI6 (interference of sleep problem with daily functioning and noticeability of impaired quality of life) and ISI2—ISI3 (difficulty staying sleep and problems waking up too early). ISI items that had significantly stronger connections with SAVE-6 and PHQ-9 were ISI7 connected to SAVE-6 and ISI2 connected to PHQ-9.

## 4. Discussion

In this study, we aimed to investigate the relationships between insomnia symptoms to identify central symptoms of insomnia in the general population using network analysis. Our secondary aim was to see how these insomnia symptoms would interact given the current COVID-19 pandemic situation by incorporating anxiety response to COVID-19 and depression severity. The results showed that ISI7 (worry about current sleep pattern) had the highest closeness and betweenness centralities, as well as fairly high strength centrality. In addition, ISI7 also had the strongest inter-scale edge weight, connected to SAVE-6 total score. On the other hand, ISI2 (difficulty staying asleep) had the strongest edge with PHQ-9 total score. These results have multiple implications regarding causal models of insomnia and their treatments.

The main finding of this study is high centrality values of ISI7. ISI7 differs from most other items of ISI that attempt to measure the severity of sleep problems themselves. Instead, ISI7 is a measure of cognitive aspect of insomnia, measuring how severely one worries about their sleep problems rather than the sleep problems themselves. High closeness and betweenness indicate widespread effect and significant bridging role of ISI7. This result is notable because it supports the role of dysfunctional beliefs towards insomnia symptoms as perpetuating factor of insomnia. Etiological models of insomnia have considered cognitive symptoms, such as dysfunctional belief, to be important in terms of development of insomnia disorder. For example, the microanalytic model [37], the cognitive model [38], and the psychobiologic inhibition model [39] of insomnia posit that acute insomnia may be perpetuated into chronic insomnia via worry and rumination about factors related to sleep, such as sleep timing, duration, and quality. Indeed, participants with insomnia complaints endorsed more dysfunctional, negative beliefs about consequences of poor sleep [40]. As dysfunctional cognitions are a key factor that reinforce exacerbation of insomnia, they are a key target for the Cognitive Behavioral Therapy for Insomnia (CBT-I), which aims to correct the maladaptive beliefs and attitudes about sleep [41,42]. In addition, addressing dysfunctional cognitions about sleep is critical to reducing subjective dissatisfaction about sleep that contributes to exacerbation of insomnia. This is also consistent in our results, as shown by the high centrality index of ISI4 (satisfaction with current sleep pattern) and edge between ISI4 and ISI7.

With regard to anxiety response to COVID-19, the edge between SAVE-6 and ISI7 was revealed to be the strongest inter-scale edge. This is a notable result that provides an important hypothesis on how the COVID-19 pandemic increases symptoms of insomnia. In a pandemic situation, excessive worries, such as worry about being infected, loss of health due to infection, or social isolation often draw more serious mental health issues, especially in a non-infected general population. Such worry can easily transpose to worry about other areas, such as worry about one’s sleep patterns as shown in our results. The important point here is that anxiety response to COVID-19 was related to worry about insomnia symptoms instead of insomnia symptoms themselves such as short sleep duration. Too much worry about sleep pattern acts as an important mechanism in the exacerbation of insomnia, as described by the etiological models of insomnia [37,38,39]. In addition, it is interesting to note that worry about COVID-19 and worry about sleep patterns both classify was repetitive negative thinking (RNT) [43]. Increasing reports have suggested RNT to have a transdiagnostic property present in multiple mental disorders, including depression, anxiety disorders, and insomnia, as suggested from etiological models of insomnia [37,38,39,44]. In this regard, our results support the relevance of RNT in insomnia, depression, and anxiety about COVID-19. These results highlight the importance of sleep education to prevent the occurrence, exacerbation, and chronicity of insomnia under the viral pandemic situation.

In our results, depression was significantly related to ISI2 (difficulty staying asleep). It is known that there is a bidirectional relationship between insomnia and depression [45,46]. Insufficient sleep may be a risk factor of depression and vice versa. However, it has not been established how each insomnia symptom differ in their effect on the onset and exacerbation of depression. Despite the cross-sectional nature of this study, our results indicate difficulty maintaining sleep to be most strongly related to depression symptoms. Our results are consistent with some previous studies showing association of depression with sleep maintenance problems [47,48]. Difficulties in maintaining sleep can be manifested by changes in sleep architecture observed in patients with depression, such as decreased slow wave sleep or increased rapid eye movement sleep [49,50]. In this regard, interventions to improve sleep maintenance may be useful in prevention of depression symptoms during pandemic situation. However, it should be noted that depression is often reported to be associated with early morning awakenings [51,52]. In our study, the edge between ISI3 (problems waking up too early) and PHQ-9 was not strong. This can be attributed to our sample being taken from the general population rather than depression patients. In addition, such discrepancy also warrants the necessity to investigate specific types of insomnia symptoms that are more strongly associated with depression. It should be noted that, in our study, PHQ-9 scores were relative to our sample taken from the general population. Therefore, future studies may benefit from conducting studies on clinical population and categorization of insomnia symptoms.

The limitations of this study are as follows. First, the result interpretations are limited by potential bidirectionality of the effects due to the cross-sectional design. Future analyses may employ directed networks such as directed acyclic graphs using longitudinal data in order to establish causal pathways between the symptoms [53]. Second, our network showed overall a modest link between insomnia symptoms, depression, and anxiety response to COVID-19. This is likely due to analyzing data from a non-psychiatric sample. Future studies may yield meaningful results from employing clinical samples. For example, one could investigate whether patients with depression experienced worsened insomnia symptoms compared to pre-pandemic symptoms.

In conclusion, the results of this study emphasized the significance of cognitive aspects of sleep within the network of insomnia symptoms. This result supports the core role played by dysfunctional cognitions in etiological models of insomnia as well as the efficacy of the popularly practiced treatment of insomnia, CBT-I. In addition, we found that anxiety response to COVID-19 was significantly related to worry about sleep patterns. Finally, we found a significant link between depression and difficulty staying asleep. This result, with regard to previous studies showing both consistency and discrepancy, necessitates the need to investigate the particular insomnia symptoms linked to depression.

## Figures and Tables

**Figure 1 jcm-11-03416-f001:**
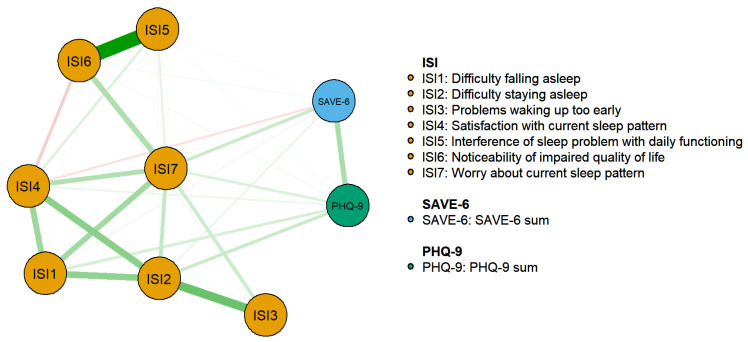
Network structure of insomnia symptoms, stress response to COVID-19, and depression level in a non-infected general population.

**Figure 2 jcm-11-03416-f002:**
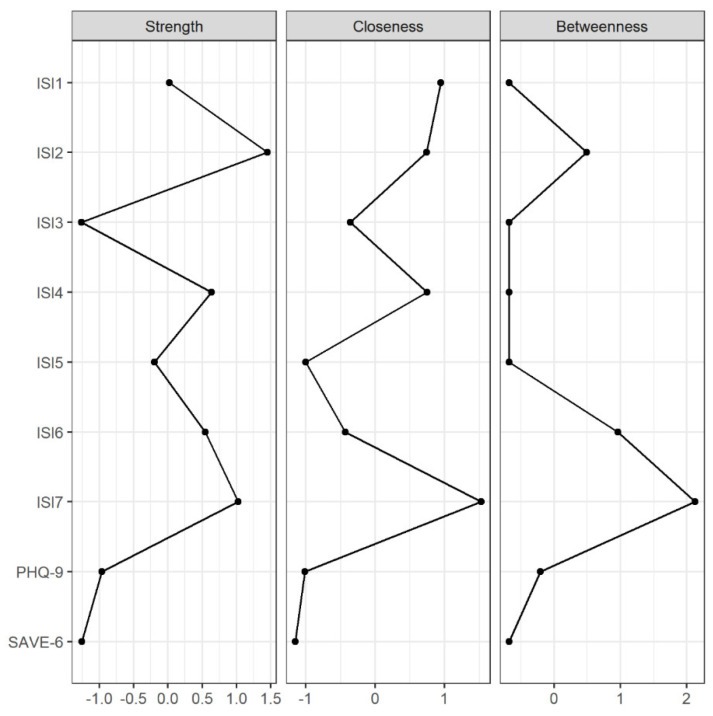
Strength, closeness, and betweenness centralities of each node of the network.

**Figure 3 jcm-11-03416-f003:**
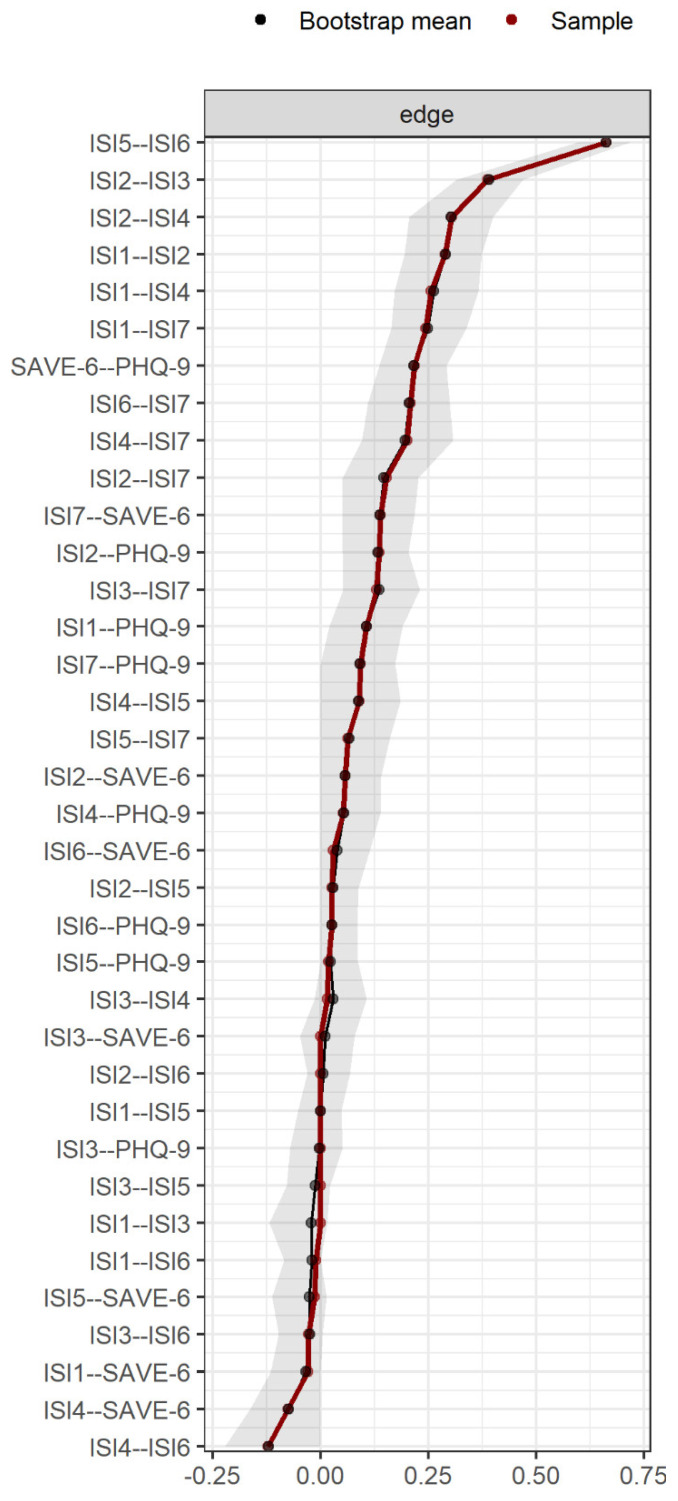
Bootstrapped edge-weights for each edge. Note. Edge-weights are ordered from strong to weak, top to bottom. Grey area indicates the bootstrapped confidence intervals for each edge.

**Figure 4 jcm-11-03416-f004:**
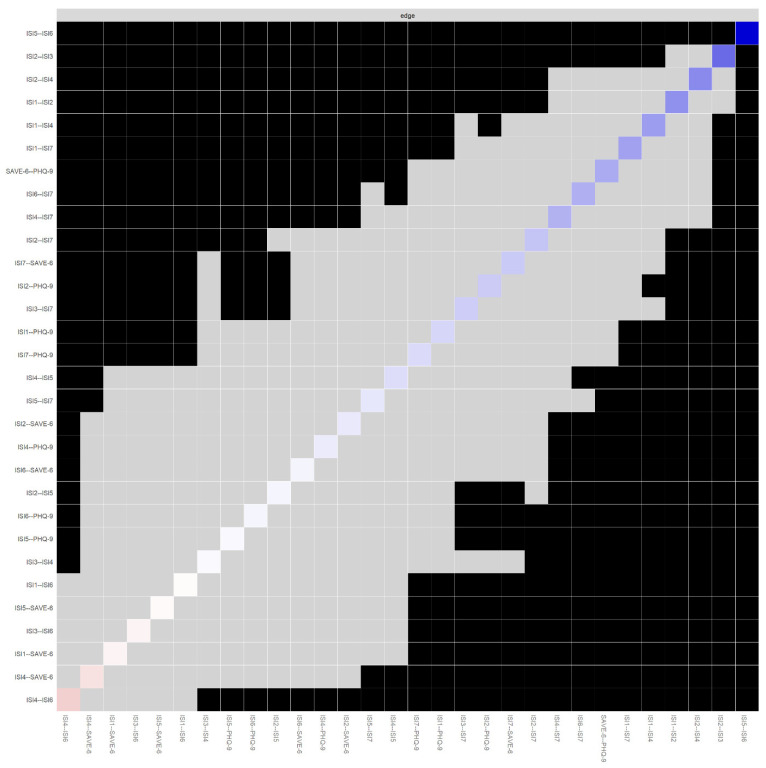
Results for bootstrapped difference tests of all edges, ordered by weight. Note. Black boxes indicate significant difference, while gray boxes indicate non-significant difference between edges located on the corresponding x and y-axis. The color of diagonal boxes indicate the strength and direction of its corresponding edge. Blue indicates positive and red indicates negative edge. Darker color indicates stronger edge. More numbers of black boxes with regard to an edge indicate that it is significantly stronger than most other edges.

**Table 1 jcm-11-03416-t001:** Table showing gender ratio, means, and standard deviations of ISI, SAVE-6, and PHQ-9, and number of participants in scale quartiles.

Variable	Value
Gender	
Males (%)	400 (51.0%)
Females (%)	385 (49.0%)
Mean age (*SD*)	43.20 (12.11)
20 to 29 (%)	154 (19.6%)
30 to 39 (%)	156 (19.9%)
40 to 49 (%)	192 (24.5%)
Above 50 (%)	283 (36.0%)
ISI mean (*SD*)	11.06 (5.39)
SAVE-6 mean (*SD*)	14.96 (4.31)
12 or below	206 (26.2%)
13 to 15	205 (26.1%)
16 to 18	231 (29.4%)
19 or above	143 (18.2%)
PHQ-9 mean (*SD*)	5.59 (5.63)
1 or below	224 (28.5%)
2 to 4	207 (26.4%)
5 to 9	191 (24.3%)
10 or above	163 (20.8%)

Note. *SD* = standard deviation; ISI = Insomnia Severity Index; SAVE-6 = Stress and Anxiety to Viral Epidemics—6; PHQ-9 = Patient Health Questionnaire—9.

## Data Availability

The data presented in this study are available in Appendix A of this article.

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
