# Peer review of "Central Symptoms of Insomnia in Relation to Depression and COVID-19 Anxiety in General Population: A Network Analysis"

_jcm, 2022, doi:10.3390/jcm11123416_

Round 1

Reviewer 1 Report

The article is interesting, well-referenced and well-grounded. Network analysis is a promising methodology, and its application might contribute significantly to understanding insomnia and improving current treatment solutions.

Overall, recommend minor revisions.

Below I will report some observations and questions for the authors with respect to each section of the paper.

[Introduction] The literature review is concise but thorough, and state of the art is successfully conveyed in this section. I appreciated the authors' attempt to introduce network analysis. Although it could have been presented in a more in-depth way, I understand the authors' choice to be as brief as possible given the clinical vocation of the journal. However, the clinical relevance of some network indexes (e.g., symptom with higher closeness = advisable intervention target) might have been explained with some examples from clinical practice or previous clinical research to make it more understandable.

[Methods] The research procedure and the measures employed are clearly described. Total scores of the SAVE-6 and the PHQ-9 were converted into ordinal scores (range 1-4) considering sample-derived quartiles for network analysis. In contrast, the items of the ISI were considered as separated nodes in the network. I am curious about the impact of this choice on the results. My argument is logical rather than statistical: ISI items are ordinal measures (range 0-4) representing insomnia symptoms and were likely randomly distributed in the sample. SAVE-6 and PHQ-9 aggregated ordinal scores, on the other hand, were "centered" on the sample distribution of the respective scales' total scores. Thus a high score on a single ISI item represented an (supposed) absolute high intensity of the specific insomnia symptom, while a high score on the ordinal PHQ-9 represented a relative high intensity of depression. I wonder whether the authors considered using conventional score ranges to discretize PHQ and SAVE scores rather than ranges inferred from the investigated sample.

Furthermore, the meaning of each node is different in this way (ISI item = symptom intensity, PHQ-9 = disorder intensity).

[Results] As a whole, results are presented in a tidy way. However, Figure 4 is not easy to understand, and a clear explanation of the bootstrapped difference test is not presented in the text.

[Discussion] Implications of the observed findings were exhaustively exposed and discussed with appropriate references to the current literature on insomnia, depression, and psychological consequences of the covid-19 pandemic. The reported centrality of the worry aspect of insomnia and its relationship with depression and anxiety is particularly interesting and recalls the relevance of transdiagnostic constructs such as repetitive negative thinking (RNT; e.g., McEvoy et al., 2010). The role of RNT in insomnia could be further explored given its possible relevance in the discussed association between insomnia, depression, and anxiety.

I appreciated that the author addressed the specific impact of CBT-I on worry highlighting the modular structure of this type of treatment, and also their effort to identify the main limitations of their study. However, with respect to this section, it is not clear to me why the impact of pre-pandemic sleep quality should be considered as a limitation (lines 218-220).

Reviewer 2 Report

The authors performed and interesting study based on questionnaires evaluating insomnia in general population, especially in covid era.

Minor concernings to improve the overall quality:

Introduction

In an epidemiology study done in Europe, this prevalence was reported to be as common as 27.2% . In addition, one study reported that 28% of insomnia patients also had current diagnosis of mental disorder. It will be very effective in terms of the quality of life of patients determine the conditions that disrupt sleep hygiene and to perform the necessary interventions. Please discuss and cite doi:10.1111/ijcp.13786

- The necessary usage of protective devices such as face mask lead to minor respiratory symtoms, insomnia, headache, esacerbating especially in healthcare workers. please discuss and cite doi:10.7416/ai.2021.2439

Results

- report al the sd with mean ± sd

- all the coefficient found in the results should also be discussed and reported in the discussion.

Author Response

Thank you for the kind comment regarding our paper.

We have accept all the points to improve the quality of our paper.

Reviewer 3 Report

The manuscript of Cha et al assesses the insomnia symptoms in relation to COVID-19 related anxiety and depression using network analysis. The data analysed was gathered in 2 online surveys and included information collected from persons without any personal history of SARS-COV2 infection. Based on the responses, the authors showed that anxiety of getting COVID-19 translated into anxiety about sleep patterns and that symptoms of depression were strongly associated with difficulty of staying asleep. This analysis provides useful insights on the effect of pandemic/quarantine on sleep patterns and mental health.

The research is well designed, the statistical analysis is appropriate and the manuscript is very well written. The limitations are mentioned by the authors in the manuscript and the originality resides from the network analysis that allowed the identification of central symptoms of insomnia and their associations in the current pandemic situation.

Given these, I do not have any comments.

Author Response

We thank you for the thorough review of our paper and the kind comments provided.